# The Relationship between the Brain-Derived Neurotrophic Factor Gene Polymorphism (Val66Met) and Substance Use Disorder and Relapse

**DOI:** 10.3390/ijms25020788

**Published:** 2024-01-08

**Authors:** Aleksandra Strońska-Pluta, Aleksandra Suchanecka, Krzysztof Chmielowiec, Jolanta Chmielowiec, Agnieszka Boroń, Jolanta Masiak, Olimpia Sipak-Szmigiel, Remigiusz Recław, Anna Grzywacz

**Affiliations:** 1Independent Laboratory of Health Promotion, Pomeranian Medical University in Szczecin, Powstańców Wielkopolskich 72 St., 70-111 Szczecin, Poland; aleksandra.stronska@pum.edu.pl (A.S.-P.); o.suchanecka@gmail.com (A.S.); 2Department of Hygiene and Epidemiology, Collegium Medicum, University of Zielona Góra, 28 Zyty St., 65-046 Zielona Góra, Poland; chmiele@vp.pl (K.C.); chmiele1@o2.pl (J.C.); 3Department of Clinical and Molecular Biochemistry, Pomeranian Medical University in Szczecin, Aleja Powstańców Wielkopolskich 72 St., 70-111 Szczecin, Poland; agnieszka.boron@pum.edu.pl; 4Second Department of Psychiatry and Psychiatric Rehabilitation, Medical University of Lublin, 1 Głuska St., 20-059 Lublin, Poland; jolanta.masiak@umlub.pl; 5Department of Obstetrics and Pathology of Pregnancy, Pomeranian Medical University, 48 Żołnierska St., 71-210 Szczecin, Poland; olimpiasipak-szmigiel@wp.pl; 6Foundation Strong in the Spirit, 60 Sienkiewicza St., 90-058 Łódź, Poland; health@mocniwduchu.pl

**Keywords:** BDNF, addiction, polymorphism

## Abstract

Substance addiction is a neuropsychiatric disorder characterized by a recurring desire to continue using a substance despite harmful consequences. Brain-derived neurotrophic factor (BDNF) is a protein that plays a role in the activity-dependent remodeling of neural function in adult nervous systems. This study analyzed the association of the rs6265 polymorphism of the BDNF gene in a group of patients addicted to psychoactive substances who were participating in addiction treatment for the first time, in a group of post-relapse psychoactive substance abusers and in a control group. The study also assessed personality and anxiety in all study groups. Statistically significant differences in the frequency of genotypes and alleles were found between all study groups. Compared to the control, both study groups had statistically significantly higher scores for trait and state anxiety. Addicted patients in both groups also had higher scores on the Neuroticism and Openness scales and lower scores on the Extraversion and Agreeableness scales. The results of this study provide further evidence that personality traits, anxiety and the rs6265 polymorphism of the BDNF gene may be risk factors for susceptibility to addiction to psychoactive substances. In addition, they can be a predictor of addiction relapse, but further extensive studies are required to confirm these findings.

## 1. Introduction

Substance addiction is a neuropsychiatric disorder characterised by a recurring desire to continue using a substance despite harmful consequences [1]. This drug-seeking behaviour is associated with craving and loss of control. The actions of drug abuse cause addiction and generally require repeated drug exposure. Both the individual’s genetic make-up and the psychological and social context in which drug use takes place strongly influence this process [2].

Concrete diagnostic criteria for substance addiction are set in DMS-V (Diagnostic and Statistical Manual of Mental Disorders) or ICD-10 (International Classification of Diseases and Related Health Problems) and have been widely used to diagnose addiction and evaluate its treatment [3]. A return to drug use after prolonged periods of abstinence is one of the most critical problems in the long-term treatment of drug addicts [4,5].

Brain-derived neurotrophic factor (BDNF) belongs to a group of secreted homodimeric proteins called neurotrophins. They are widely recognised as regulators of cell growth, survival and differentiation during nervous system development. Neurotrophins also play an important role in the activity-dependent remodelling of neural function in adult nervous systems [6]. It is increasingly recognised that such activity-dependent remodelling is crucial for the transition from casual substance use to substance dependence, leading to the investigation of the role of BDNF in the actions of substance abusers. In particular, there has been considerable interest in the effects of BDNF on reward-related neuronal circuits, which include dopamine (DA) neurons of the ventral tegmental area (VTA) and related targets in the forebrain, such as the nucleus accumbens (NAc) and prefrontal cortex (PFC) [7].

There is an urgent need to understand the mechanisms underlying relapse to substance use. Several factors have already been shown to play a role in relapse susceptibility, including dysregulation of the hypothalamic–pituitary–adrenal axis and overactivity of the CRF system [8,9], a high cortisol/ACTH ratios [10], a lower amygdala volume [11], or depressive symptoms [12,13]. Other studies have also suggested that higher serum brain-derived neurotrophic factor (BDNF) levels are highly predictive of addiction relapse, at least in cocaine abusers [14].

A common single-nucleotide polymorphism has been identified in the human BDNF gene. This polymorphism involves an amino acid substitution from valine to methionine at the 66th amino acid residue (Val66Met). This genetic variation is associated with altering intracellular trafficking and packaging of proBDNF and, consequently, with the secretion of mBDNF [15]. The Val66Met BDNF gene polymorphism has been linked with decreased BDNF serum concentrations [16].

The five-factor model of personality has been proposed as a classification system for people who are prone to addictive behaviour. This model describes five basic personality dimensions: Neuroticism, Conscientiousness, Extraversion, Agreeableness, and Openness to Experience, which encompasses an individual’s behavioural, emotional, and cognitive patterns [17].

This study aimed to analyse the association of the rs6265 polymorphism of the BDNF gene in a group of patients addicted to psychoactive substances participating in addiction treatment for the first time, in a group of post-relapse psychoactive substance abusers and a control group. The study also assessed personality measured using the NEO-FFI questionnaire and anxiety measured using the STAI questionnaire in all study groups.

## 2. Results

The frequency distributions concurred with the Hardy–Weinberg equilibrium (HWE) both in a group of patients addicted to psychoactive substances participating in addiction treatment for the first time, in a group of patients addicted to psychoactive substances after relapse and in the control group (Table 1).

Table 2 shows the frequency of genotypes and alleles of the rs6265 polymorphism in the BDNF gene in all studied groups.

The means and standard deviations of the NEO-FFI results and the STAI state and trait scales for the patients addicted to psychoactive substances participating in addiction treatment for the first time, patients addicted to psychoactive substances after relapse and the control group are shown in Table 3.

### 2.1. Patients in Addiction Treatment for the First Time vs. the Control Group

The frequencies of the rs6265 BDNF genotypes were statistically significantly different in the tested sample of patients in addiction treatment for the first time when compared to the control group (G/G 0.71 vs. G/G 0.59; A/A 0.02 vs. A/A 0.06; A/G 0.27 vs. A/G 0.35, χ^2^ = 6.376, *p* = 0.0412) (Table 2).

Statistically significant differences in the frequency of rs6265 BDNF alleles were found between patients in addiction treatment for the first time and in the control group (G 0.84 vs. G 0.77; A 0.16 vs. A 0.23, χ^2^ = 6.390, *p* = 0.0115) (Table 2).

Patients in addiction treatment for the first time compared to the control group scored higher in assessing the STAI trait scale (6.94 vs. 5.18; Z = 6.622; *p* < 0.0001) STAI state scale (5.58 vs. 4.72; Z = 3.549; *p* = 0.0003), NEO-FFI Neuroticism (6.42 vs. 4.71; Z = 6.806; *p* < 0.0001) and NEO-FFI Openness scales (5.13 vs. 4.59; Z = 2.341; *p* = 0.0192). Significantly lower results were obtained for the NEO-FFI Extraversion scale (5.88 vs. 6.39; Z = −2.266; *p* = 0.0234) and the Agreeableness scale (4.53 vs. 5.43; Z = −3.995; *p* = 0.0001) (Table 3).

The results of the 2 × 3 factorial ANOVA of the NEO Five-Factor Personality Inventory and the State-Trait Anxiety Inventory sten scales are summarised in Table 4.

A significant statistical impact of first-time therapy and rs6265 BDNF genotype was demonstrated for the NEO-FFI Openness scale score.

There was a statistically significant effect of rs6265 BDNF genotype interaction for first-time treatment or not addicted (control) groups on the Openness scale (F_2.378_ = 7.68; *p* = 0.0005; η^2^ = 0.039; Figure 1). The power observed for this factor was 95%, and approximately 4% was explained by the polymorphism of the rs6265 BDNF and first-time therapy or lack thereof on trait Openness score variance. There was also a statistically significant effect of the rs6265 BDNF genotype on the Openness scale score (F_2.378_ = 8.29; *p =* 0.0003; η^2^ = 0.042). The power observed for this factor was over 96%, and approximately 4% was explained by first-time therapy or lack thereof on the variance in the Openness score. The results of the post hoc test are shown in Table 6.

### 2.2. Patients with Addiction Relapse vs. the Control Group

The rs6265 BDNF genotype was significantly different in relapsed patients compared to controls. (G/G 0.72 vs. G/G 0.59; A/A 0.02 vs. A/A 0.06; A/G 0.26 vs. A/G 0.35, χ^2^ = 8.074, *p* = 0.01765) (Table 2).

Statistically significant differences in the frequency of rs6265 BDNF alleles were found between patients with addiction relapse and the control group (G 0.85 vs. G 0.77; A 0.15 vs. A 0.23, χ^2^ = 8.220, *p* = 0.0041) (Table 2).

The group of patients with a relapse of addiction in comparison with the control group achieved higher scores in the assessment of the STAI trait scale (7.30 vs. 5.19; Z = 8.081; *p* < 0.0001) STAI state scale (6.16 vs. 4.72; Z = 5.840; *p* < 0.0001), NEO-FFI Neuroticism (7.01 vs. 4.71; Z = 9.489; *p* < 0.0001) and NEO-FFI Openness scales (5.24 vs. 4.58; Z = 3.170; *p* = 0.0015). Significantly lower results were obtained for the NEO-FFI Extraversion (5.62 vs. 6.39; Z = −3.294; *p* = 0.0009) and the Agreeableness (4.06 vs. 5.43; Z = −6.305; *p* < 0.0001), and the Conscientiousness scales (5.43 vs. 5.93; Z = −2.125; *p* = 0.0335) (Table 3).

The results of the 2 × 3 factorial ANOVA of the NEO Five-Factor Personality Inventory and the State-Trait Anxiety Inventory sten scales are summarised in Table 5.

A significant statistical impact of addiction relapse and rs6265 BDNF genotype was demonstrated for the STAI trait scale score.

There was a statistically significant effect of rs6265 BDNF genotype interaction for patients with addiction relapse or those who were not using (control group) on the STAI trait scale (F_2.384_ = 5.03; *p* = 0.0070; η^2^ = 0.026; Figure 2). The power observed for this factor was 82%, and approximately 3% was explained by the polymorphism of the rs6265 BDNF and relapse or being in the control group on trait STAI trait scale variance. There was also a statistically significant effect of relapse or not using (the control group) on the STAI trait scale score (F_1.384_ = 3.95; *p =* 0.0474; η^2^ = 0.010). The power observed for this factor was over 51%, and approximately 1% was explained by relapse or lack thereof on the variance in the STAI trait scale. Table 6 shows the results of the post hoc test.

### 2.3. Patients in Addiction Treatment for the First Time vs. Patients with Addiction Relapse

The group of patients in addiction treatment for the first time compared to the group of patients with addiction relapse obtained higher scores in the assessment of Agreeableness (4.53 vs. 4.06; Z = 2.060; *p* = 0.0393). Significantly lower results were obtained for the STAI trait scale (6.94 vs. 7.30; Z = −1.959; *p* = 0.0500) and the Neuroticism scale (6.42 vs. 7.01; Z = −2.527; *p* = 0.0115).

## 3. Discussion

BDNF has been linked to many mental disorders. The rs6265 polymorphism of the *BDNF* gene causing the replacement of the valine amino acid at 66^TH^ position with methionine in the BDNF propeptide does not affect the sequence of the mature BDNF protein. Still, it leads to reduced distribution of BDNF to the dendrites of neurons and reduced activity-dependent secretion [18,19]. People with the Met allele had smaller hippocampal volumes [20,21] and exhibited poorer performance during hippocampus-dependent memory tasks [15]. Therefore, the Met allele, which is responsible for reduced BDNF availability, is considered by many researchers to be a risk factor for psychopathology, including increased neuroticism, anxiety, depression, and suicide [22,23,24,25].

Our study aimed to analyse the relationship between the rs6265 polymorphism of the *BDNF* gene and addiction to psychoactive substances in patients starting addiction treatment for the first time and in patients with addiction relapse, as well as to analyse personality traits assessed using the NEO-FFI Inventory; anxiety measured using the STAI questionnaire; and finally, the interaction between rs6265 and personality traits and anxiety.

The results of our association study indicate statistically significant differences in genotype frequencies. The GG genotype occurred statistically significantly more often in the group of addicted patients receiving addiction treatment for the first time and in the group of patients with addiction relapse compared to the control group. Our research also shows a significantly different distribution of rs6265 alleles. The G allele occurred significantly more often in the group of addicted patients undergoing addiction treatment for the first time and in the group of patients with addiction relapse compared to the control group.

Cheng et al. [26] conducted an association study of the rs6265 polymorphism of the *BDNF* gene in a group of methamphetamine and heroin addicts and the control group. They found significant differences in the distribution of the *BDNF* Val66Met genotype between people addicted to methamphetamine or heroin and the control group, which suggests that a lower frequency of the 66Met carrier is associated with substance abuse, which is also confirmed by studies by Jia et al. [27], who also analysed whether the rs6265 polymorphism of the *BDNF* gene is associated with heroin addiction. The results of their study showed that the G allele of rs6265 was significantly more frequent in heroin addicts than in controls. Their results are consistent with those obtained in this study.

Different results were obtained by Meng et al. [28]. The allele distribution of the rs6265 polymorphism of the *BDNF* gene did not differ significantly between heroin-addicted patients and the control group in their study. They noted, however, that AA *BDNF* rs6265 carriers had an earlier onset of heroin addiction and a more pronounced trend in a family history of heroin addiction than GG carriers after controlling for behavioural characteristics within rs6265 genotypes.

In a group of people abusing methamphetamine, Su et al. [29] investigated the relationship between impulsivity and the Val66Met polymorphism of the BDNF gene. They assessed impulsivity using the Chinese version of the Barratt Impulsivity Scale-11 (BIS-11). They also examined the association of the polymorphism with the age of onset of methamphetamine abuse. Their results showed that there were no significant differences in the distribution of genotypes and alleles between the methamphetamine users and the control group. In the methamphetamine abuse group, people carrying the Met allele had significantly higher BIS scores for attentional impulsivity compared to people with the Val/Val genotype. An earlier onset of methamphetamine use was also associated with the Met allele.

Addiction is a multifactorial disease. In addition to genetic factors, personality traits may also contribute to the use of psychoactive substances [30]. The five-factor model of personality, also known as the Big Five, assumes that personality consists of five basic dimensions: Neuroticism, Agreeableness, Openness, Conscientiousness and Extraversion. Each dimension describes enduring character traits influencing people’s thoughts, feelings and behaviour. The Big Five Theory was developed based on lexical research analysing the words used to describe personality in different languages. Researchers used a statistical method called factor analysis to find common factors or dimensions underlying these words. Such an analysis was conducted for the first time by Fiske [31] in 1949, and later his results were confirmed and developed by other scientists [32,33,34,35]. Factor analysis showed that the five personality dimensions exist across cultures and contexts [36] and among people with mental disorders [37]. There is no clear definition of each dimension, but they can be described as follows: Neuroticism is the tendency to experience negative emotions like fear, sadness or anger. Agreeableness is the tendency to cooperate, be kind and care for others. Openness is the tendency to seek new experiences, be creative and be curious about the world. Conscientiousness is the tendency be organised, responsible and hard-working. Extraversion is the tendency to be sociable, energetic and assertive [38].

In our study, personality was assessed using the NEO-FFI Inventory. The analysis showed that patients participating in addiction treatment for the first time, compared to the control group, obtained higher scores on the Neuroticism and Openness scales. The same was true when comparing the group of patients with addiction relapse and the control group. However, when comparing the group of patients participating in addiction treatment for the first time with the group of patients with addiction relapse, patients participating in addiction treatment for the first time scored statistically significantly lower on the Neuroticism scale. On the Extraversion and Agreeableness scales, patients participating in addiction treatment for the first time and patients with addiction relapse achieved statistically significantly lower results compared to the control group. However, when comparing the group of patients participating in addiction treatment for the first time with the group of patients with addiction relapse, patients participating in addiction treatment for the first time had a statistically significantly lower result on the Agreeableness scale.

In this study, we also assessed the impact of the rs6265 genotype interaction of the BDNF gene in groups of people with a substance use disorder in treatment for the first time, as well as its impact in those experiencing addiction relapse and those in the control group. When comparing the group of patients participating in addiction treatment for the first time with the control group, a statistically significant impact of the interaction of the rs6265 genotype of the BDNF gene and first-time addiction treatment or no drug use (control group) on the Openness scale was found. A statistically significant effect of the rs6265 genotype of the BDNF gene on the value result on the Openness scale was also found.

Kulkarni et al. [39] conducted a study to assess the level of nicotine addiction in tobacco smokers (working in the corporate sector) to learn about their personality profiles and the relationship between personality traits and continued smoking. They showed that Neuroticism was significantly related to the level of nicotine addiction. Extraversion and openness were associated with health problems, while agreeableness and conscientiousness were related to social factors as a reason for quitting smoking. Extraversion and Agreeableness have been linked to occupational and social factors as causes of relapse. Indeed, a growing body of literature examines the links between the Big Five and illicit drug use [40,41,42]. Research conducted by Terracciano et al. [41] among 1102 adults allowed us to conclude that people with a high level of Neuroticism were more likely to use cannabis. Moreover, people with lower Agreeableness and Conscientiousness scores were more likely to use cannabis. Similar results were obtained by Dash et al. [40], who examined a sample of 3785 twins and siblings from Australia. They showed that high levels of Neuroticism, low Agreeableness and Conscientiousness were associated with cannabis addiction. Similar results were also obtained by Sutin et al. [42] on a sample of 412 adults, which showed that higher levels of Neuroticism and lower scores for Agreeableness and Conscientiousness were associated with a greater likelihood of using cocaine or opiates.

The *BDNF* gene may be associated with the putative common pathophysiology of depression and anxiety. One common, non-conservative polymorphism has been identified in the human *BDNF* gene, rs6265, which causes substitution at codon 66. Several studies suggest that the Met allele is associated with reduced hippocampal volume and abnormal hippocampal activation in humans [15,43,44], and the hippocampus plays a role in regulating mood state. It can be expected that the *BDNF* Val66Met polymorphism may influence behaviour and anxiety.

Anxiety disorders often co-exist with substance dependence and are more common in families with a history of using psychoactive substances [45].

Using the STAI questionnaire, our study also analysed anxiety as a trait and state. Compared to the control group, patients participating in addiction treatment for the first time achieved statistically significantly higher results in the assessment of anxiety as a trait and as a state. Patients with addiction relapses, compared to the control group, also obtained statistically significantly higher results of anxiety as a trait and as a state. When comparing the group of patients participating in addiction treatment for the first time with the group of patients with addiction relapse, patients participating in addiction treatment for the first time obtained a statistically significantly lower score on the STAI trait scale.

We also assessed the interactions of the rs6265 genotype of the *BDNF* gene and relapse or not using (the control group) with the STAI trait scale. We also found a statistically significant impact of addiction relapse or not using (the control group) on the STAI trait score.

Pietras et al. [46] conducted a study that assessed anxiety as a trait and as a state measured using the State-Trait Anxiety Inventory (STAI) among nicotine addicts and the control group. They showed that the average intensity of trait anxiety and state anxiety and its level differ between groups.

Despite many studies on the influence of the rs6265 polymorphism of the BDNF gene and susceptibility to addiction, it is not easy to find a consensus in this case. This may be explained by various factors, including study samples that are generally relatively small and effects that may be specific to gender and ethnicity.

Addictions are multifactorial and polygenic. Therefore, there is still a need for research on the association of candidate gene polymorphisms because although a single polymorphism has a negligible impact on the manifestation of the addiction phenotype, they are still unrecognised.

## 4. Materials and Methods

### 4.1. Participants

The study group consisted of 533 volunteers. Of these, 291 were addicted to psychoactive substances (mean age = 28.87, SD = 6.58) and 242 were in the control group (mean age = 22.52, SD = 4.14). The Pomeranian Medical University Bioethics Committee in Szczecin approved the study (KB-006/01/2022). Before participation in the study, all participants gave written informed consent. The study was conducted in the Independent Health Promotion Laboratory, Pomeranian Medical in Szczecin. Both participants who were addicted to psychoactive substances and the control group participants underwent the same psychometric testing with the State-Trait Anxiety Inventory (STAI) questionnaire and NEO Five-Factor Personality Inventory (NEO-FFI). The study group included people participating in addiction treatment for the first time (143) and people experiencing addiction relapse (148).

### 4.2. Psychometric Tests

The STAI questionnaire is a tool that measures anxiety as a trait, which is a persistent predisposition to having worries, stress, discomfort, and anxiety. It also measures anxiety as a state, which takes the form of fear, anxiety, and momentary stimulation of the autonomic nervous system in response to specific situations.

For each of the five traits, the NEO-FFI Five-Factor Inventory contains six components—Neuroticism (anxiety, hostility, depression, self-consciousness, impulsivity and susceptibility to stress), Extroversion (warmth, sociability, assertiveness, activity, emotion seeking and positive emotions), Agreeableness (trust, straightforwardness, altruism, humility and openness to experience), and positive emotions), openness to experience (fantasy, aesthetics, feelings, actions, ideas and values), agreeableness (trust, straightforwardness, altruism, compliance, modesty and tenderness) and conscientiousness (competence, orderliness, duty, striving for achievement, self-discipline and consideration) [47].

The results of both inventories, i.e., STAI and NEO-FFI, were reported as the sten scores. The conversion of the raw score to the sten scale was carried out according to the Polish standards for adults, where it was assumed that 1–2 sten corresponds to very low results, 3–4 corresponds to low results, 5–6 corresponds to average results, 7–8 corresponds high results and 9–10 sten corresponds to very high results.

### 4.3. Genotyping

The genomic DNA was isolated from venous blood using standard procedures. Genotyping was performed using the real-time PCR method. The fluorescence signal was plotted as a function of temperature to provide melting curves for each sample. The rs6265 BDNF polymorphic site peaks were read at 53.29 °C for the A allele and 59.93 °C for the G allele.

### 4.4. Statistical Analysis

The concordance between the genotype frequency distribution and Hardy–Weinberg equilibrium (HWE) was tested using the HWE software (https://wpcalc.com/en/equilibrium-hardy-weinberg/ accessed on 5 April 2023). The relations between rs6265 *BDNF* variants: first-time therapy subjects, addiction relapse subjects and control subjects and the NEO Five-Factor Inventory were analysed using a multivariate analysis of factor effects ANOVA [NEO-FFI/scale STAI/× genetic feature × control and first-time treatment, addiction relapse × (genetic feature × control and first-time treatment, relapses addiction)]. The condition of homogeneity of variance was fulfilled (Levene test *p* > 0.05). The analysed variables were not distributed normally. The NEO Five-Factor Inventory (Neuroticism, Extraversion, Openness, Agreeability and Conscientiousness) sten scores were compared using the U Mann–Whitney test. rs6265 *BDNF* genotype frequencies differences between control subjects and first-time treatment and relapse addiction were tested using the chi-square test. All computations were performed using STATISTICA 13 (Tibco Software Inc., Palo Alto, CA, USA) for Windows (Microsoft Corporation, Redmond, WA, USA). Statistical significance was assumed at *p* < 0.05.

## 5. Conclusions

The results of this study provide further evidence that personality traits, anxiety, and the rs6265 polymorphism of the *BDNF* gene may be risk factors for susceptibility to addiction to psychoactive substances. In addition, they can be a predictor of addiction relapse, but further extensive studies of this type are needed to confirm the results of this study.

We emphasise that analysing genotypes and alleles in connection with personality-related factors is justified. Still, it should be remembered that it may also be a factor limiting the interpretation of the study.

However, the innovative nature of the study makes it necessary to create homogeneous subgroups—including those that consider the respondents’ personality characteristics.

## Figures and Tables

**Figure 1 ijms-25-00788-f001:**
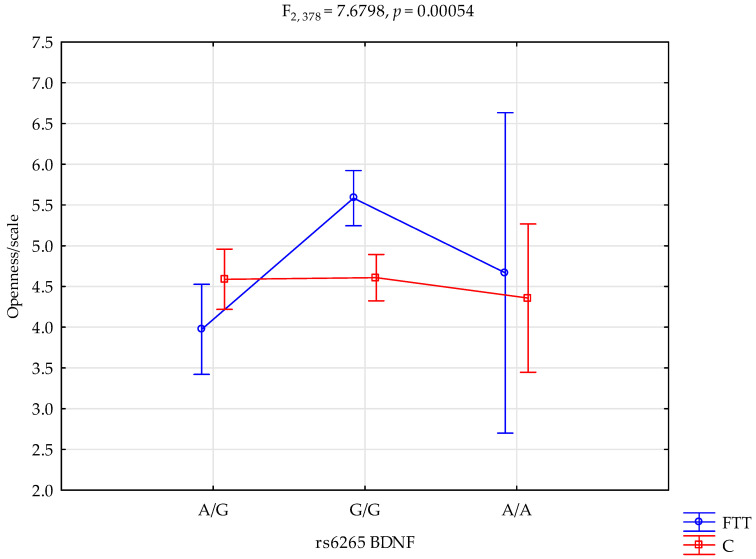
Interaction between first-time in addiction treatment (FTT)/control (C) and rs6265 *BDNF* and Openness scale.

**Figure 2 ijms-25-00788-f002:**
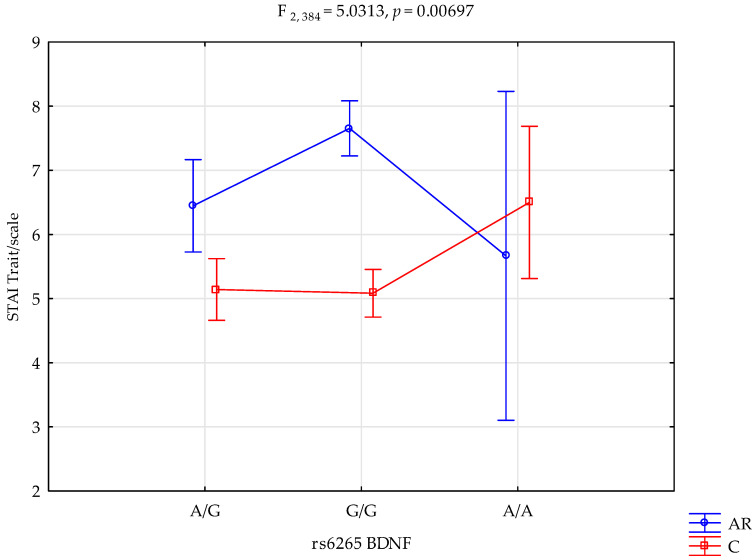
Interaction between addiction relapses (AR)/control (C) and rs6265 *BDNF* and STAI trait scale.

**Table 1 ijms-25-00788-t001:** Hardy–Weinberg’s equilibrium for rs6265 is located in the *BDNF* gene.

Hardy–Weinberg Equilibrium, Including Analysis for Estimation Bias		Observed (Expected)	Allele Freq	χ^2^ (*p*-Value)
rs6265 BDNF patients in addiction treatment for the first time *n* = 143	G/G	101 (101.5)	*p* (G) = 0.84 q (A) = 0.16	0.1161 (0.7333)
	A/A	3 (3.5)		
	A/G	39 (37.9)		
rs6265 BDNF patients with addiction relapse *n* = 148	G/G	107 (107.3)	*p* (G) = 0.85 q (A) = 0.15	0.0308 (0.8606)
	A/A	3 (3.3)		
	A/G	38 (37.5)		
rs6265 BDNF control *n* = 242	G/G	143 (142.2)	*p* (G) = 0.77 q (A) = 0.23	0.084 (0.7714)
	A/A	14 (13.2)		
	A/G	85 (86.6)		

*p*—statistical significance χ^2^ test.

**Table 2 ijms-25-00788-t002:** Frequency of genotypes and alleles of the rs6265 polymorphisms in the *BDNF* gene in the group of patients in addiction treatment for the first time, patients with relapses of addiction and the control group.

	rs6265 BDNF				
	Genotypes			Alleles	
	G/G *n* (%)	A/A *n* (%)	A/G *n* (%)	G *n* (%)	A *n* (%)
First-time treatment *n* = 143	101 (70.63%)	3 (2.10%)	39 (27.27%)	241 (84.27%)	45 (15.73%)
Control *n* = 242	143 (59.09%)	14 (5.79%)	85 (35.12%)	371 (76.65%)	113 (23.35%)
χ^2^ (*p*-value)	6.376 (0.0412) *			6.390 (0.0115) *	
Relapse addiction *n* = 148	107 (72.30%)	3 (2.03%)	38 (25.68%)	252 (85.14%)	296 (14.86%)
Control *n* = 242	143 (59.09%)	14 (5.79%)	85 (35.12%)	371 (76.65%)	113 (23.35%)
χ^2^ (*p*-value)	8.074 (0.01765) *			8.220 (0.0041) *	

*—significant statistical differences. *n*—number of subjects.

**Table 3 ijms-25-00788-t003:** STAI and NEO Five-Factor Inventory sten scores in the control group, patients in first-time treatment, and patients with relapses of addiction.

**STAI/NEO Five-Factor Inventory**	**First-Time Treatment** **(*n* = 143)**	**Control** **(*n* = 242)**	**Z**	**(*p*-Value)**
STAI trait scale	6.94 ± 2.28	5.18 ± 2.29	6.622	0.0000 *
STAI state scale	5.58 ± 2.46	4.72 ± 2.20	3.549	0.0003 *
Neuroticism scale	6.42 ± 2.16	4.71 ± 2.06	6.806	0.0000 *
Extraversion scale	5.88 ± 2.15	6.39 ± 1.99	−2.266	0.0234 *
Openness scale	5.13 ± 2.03	4.59 ± 1.61	2.341	0.0192 *
Agreeability scale	4.53 ± 1.98	5.43 ± 2.09	−3.995	0.0001 *
Conscientiousness scale	5.84 ± 2.13	5.92 ± 2.17	−0.270	0.7872
**STAI/NEO Five-Factor Inventory**	**Relapse of addiction** **(*n* = 148)**	**Control** **(*n* = 242)**	**Z**	**(*p*-Value)**
STAI trait scale	7.30 ± 2.29	5.19 ± 2.29	8.081	0.0000 *
STAI state scale	6.16 ± 2.37	4.72 ± 2.20	5.840	0.0000 *
Neuroticism scale	7.01 ± 2.12	4.71 ± 2.06	9.489	0.0000 *
Extraversion scale	5.62 ± 2.17	6.39 ± 1.99	−3.294	0.0009 *
Openness scale	5.24 ± 2.18	4.58 ± 1.61	3.170	0.0015 *
Agreeability scale	4.06 ± 1.84	5.43 ± 2.09	−6.305	0.0000 *
Conscientiousness scale	5.43 ± 2.31	5.93 ± 2.17	−2.125	0.0335 *
**STAI/NEO Five-Factor Inventory**	**First-time treatment** **(*n* = 143)**	**Relapse of addiction** **(*n* = 148)**	**Z**	**(*p*-Value)**
STAI trait scale	6.94 ± 2.28	7.30 ± 2.29	−1.959	0.0500 *
STAI state scale	5.58 ± 2.46	6.16 ± 2.37	−1.421	0.1553
Neuroticism scale	6.42 ± 2.16	7.01 ± 2.12	−2.527	0.0115 *
Extraversion scale	5.88 ± 2.15	5.62 ± 2.17	0.842	0.3995
Openness scale	5.13 ± 2.03	5.24 ± 2.18	−0.674	0.5000
Agreeability scale	4.53 ± 1.98	4.06 ± 1.84	2.060	0.0393 *
Conscientiousness scale	5.84 ± 2.13	5.43 ± 2.31	1.628	0.1035

M ± SD—mean ± standard deviation; *p*—statistical significance with Mann–Whitney U-test; *n*—number of subjects; *—statistically significant differences.

**Table 4 ijms-25-00788-t004:** The results of 2 × 3 factorial ANOVA for first-time in addiction treatment and controls, NEO Five-Factor Inventory, STAI, and *BDNF* gene rs6265.

STAI/NEO Five-Factor Inventory	Group	rs6265 BDNF				ANOVA		
		G/G *n* = 243 M ± SD	A/A *n* = 17 M ± SD	A/G *n* = 123 M ± SD	Factor	F (*p*-Value)	η^2^	Power (Alfa = 0.05)
STAI trait scale	First time in addiction treatment (FTT); *n* = 143	5.50 ± 2.53	7.33 ± 2.08	5.66 ± 2.30	Intercept FTT/control rs6265 FTT/control × rs6265	F_1.378_ = 435.92 (*p* < 0.0001) F_1.378_ = 7.64 (*p* = 0.0059) F_2.378_ = 0.76 (*p* = 0.4641) F_2.378_ = 0.93 (*p* = 0.3950)	0.536 0.020 0.004 0.005	1.000 0.788 0.181 0.211
	Control; *n* = 242	4.80 ± 2.25	4.79 ± 2.07	4.57 ± 2.18				
STAI state scale	First time in addiction treatment (FTT); *n* = 143	6.78 ± 2.37	8.67 ± 1.15	7.24 ± 2.03	Intercept FTT/control rs6265 FTT/control × rs6265	F_1.378_ = 653.12 (*p* < 0.0001) F_1.378_ = 14.93 (*p* = 0.0001) F_2.378_ = 2.76 (*p* = 0.0646) F_2.378_ = 0.31 (*p* = 0.7348)	0.634 0.038 0.014 0.002	1.000 0.971 0.543 0.099
	C: Control; *n* = 242	5.08 ± 2.22	6.50 ± 2.74	5.14 ± 2.29				
Neuroticism scale	First time in addiction treatment (FTT); *n* = 143	6.43 ± 2.24	8.33 ± 1.15	6.26 ± 1.94	Intercept FTT/control rs6265 FTT/control × rs6265	F_1.378_ = 620.51 (*p* < 0.0001) F_1.378_ = 21.66 (*p* = 0.0000) F_2.378_ = 1.73 (*p* = 0.1789) F_2.378_ = 0.68 (*p* = 0.5048)	0.621 0.054 0.009 0.004	1.000 0.996 0.362 0.165
	Control; *n* = 242	4.76 ± 2.14	5.07 ± 2.05	4.57 ± 1.92				
Extraversion scale	First time in addiction treatment (FTT); *n* = 143	5.95 ± 2.18	5.33 ± 0.58	5.73 ± 2.18	Intercept FTT/control rs6265 FTT/control × rs6265	F_1.378_ = 651.73 (*p* < 0.0001) F_1.378_ = 1.30 (*p* = 0.2557) F_2.378_ = 0.54 (*p* = 0.5847) F_2.378_ = 0.70 (*p* = 0.4979)	0.633 0.003 0.003 0.004	1.000 0.206 0.138 0.168
	Control; *n* = 242	6.30 ± 2.10	5.64 ± 1.94	6.66 ± 1.80				
Openness scale	First time in addiction treatment (FTT); *n* = 143	5.58 ± 1.99	4.67 ± 1.53	3.97 ± 1.68	Intercept FTT/control rs6265 FTT/control × rs6265	F_1.378_ = 559.45 (*p* < 0.0001) F_1.378_ = 0.32 (*p* = 0.5682) F_2.378_ = 8.29 (*p* = 0.0003) F_2.378_ = 7.68 (*p* = 0.0005 *)	0.596 0.001 0.042 0.039	1.000 0.088 0.960 0.947
	Control; *n* = 242	4.60 ± 1.59	4.36 ± 1.86	4.59 ± 1.63				
Agreeability scale	First time in addiction treatment (FTT); *n* = 1431	4.64 ± 2.06	6.00 ± 2.65	4.11 ± 1.67	Intercept FTT/control rs6265 FTT/control × rs6265	F_1.378_ = 504.71 (*p* < 0.0001) F_1.378_ = 1.61 (*p* = 0.2046) F_2.378_ = 1.69 (*p* = 0.1847) F_2.378_ = 0.94 (*p* = 0.3923)	0.572 0.004 0.009 0.005	1.000 0.244 0.356 0.212
	Control; *n* = 242	5.42 ± 2.15	5.71 ± 2.33	5.38 ± 1.96				
Conscientiousness scale	First time in addiction treatment (FTT); *n* = 143	5.76 ± 2.34	4.33 ± 0.58	6.16 ± 2.25	Intercept FTT/control rs6265 FTT/control × rs6265	F_1.378_ = 524.49 (*p* < 0.0001) F_1.378_ = 1.56 (*p* = 0.2128) F_2.378_ = 2.34 (*p* = 0.0974) F_2.378_ = 0.91 (*p* = 0.4021)	0.581 0.004 0.012 0.005	1.000 0.238 0.473 0.208
	Control; *n* = 242	5.67 ± 2.28	6.14 ± 2.51	6.31 ± 1.86				

*—significant result; FTT—first time in addiction treatment; M ± SD-mean ± standard deviation.

**Table 5 ijms-25-00788-t005:** The results of 2 × 3 factorial ANOVA for patients with addiction relapses and controls, NEO Five-Factor Inventory, STAI, and *BDNF* gene rs6265.

STAI/NEO Five-Factor Inventory	Group	rs6265				ANOVA		
		G/G *n* = 250 M ± SD	A/A *n* = 17 M ± SD	A/G *n* = 123 M ± SD	Factor	F (*p*-Value)	η^2^	Power (Alfa = 0.05)
STAI trait scale	Addiction relapses (AR); *n* = 148	7.65 ± 2.09	5.67 ± 1.53	6.44 ± 2.63	Intercept AR/control rs6265 AR/control × rs6265	F_1.384_ = 568.49 (*p* < 0.0001) F_1.384_ = 3.95 (*p* = 0.0474) F_2.384_ = 2.39 (*p* = 0.0928) F_2.384_ = 5.03 (*p* = 0.0070) *	0.597 0.010 0.012 0.026	1.000 0.509 0.482 0.815
	Control; *n* = 242	5.08 ± 2.22	6.50 ± 2.74	5.14 ± 2.28				
STAI state scale	Addiction relapses (AR); *n* = 148	6.27 ± 2.24	6.33 ± 1.53	5.84 ± 2.76	Intercept AR/control rs6265 AR/control × rs6265	F_1.384_ = 447.75 (*p* < 0.0001) F_1.384_ = 7.71 (*p* = 0.0057) F_2.384_ = 0.78 (*p* = 0.4594) F_2.384_ = 0.08 (*p* = 0.9263)	0.538 0.020 0.004 0.0004	1.000 0.791 0.183 0.061
	C: Control; *n* = 242	2.76 ± 2.24	4.78 ± 2.01	4.57 ± 2.18				
Neuroticism scale	Addiction relapses (AR); *n* = 148	7.13 ± 2.01	6.33 ± 1.15	6.71 ± 2.46	Intercept AR/control rs6265 AR/control × rs6265	F_1.384_ = 597.44 (*p* < 0.0001) F_1.384_ = 16.59 (*p* = 0.0001) F_2.384_ = 0.79 (*p* = 0.4500) F_2.384_ = 0.41 (*p* = 0.6627)	0.608 0.041 0.004 0.002	1.000 0.982 0.186 0.116
	Control; *n* = 242	4.76 ± 2.14	5.07 ± 2.06	4.57 ± 1.92				
Extraversion scale	Addiction relapses (AR); *n* = 148	5.57 ± 2.21	6.67 ± 2.31	5.68 ± 2.11	Intercept RT/control rs6265 AR/control × rs6265	F_1.384_ = 679.69 (*p* < 0.0001) F_1.384_ = 0.24 (*p* = 0.6235) F_2.384_ = 0.49 (*p* = 0.6136) F_2.384_ = 1.06 (*p* = 0.3460)	0.638 0.0006 0.002 0.005	1.000 0.077 0.130 0.236
	Control; *n* = 242	6.31 ± 2.10	5.64 ± 1.95	6.66 ± 1.80				
Openness scale	Addiction relapses (AR); *n* = 148	5.19 ± 2.19	5.67 ± 1.53	5.37 ± 2.21	Intercept RT/control rs6265 AR/control × rs6265	F_1.384_ = 560.99 (*p* < 0.0001) F_1.384_ = 4.50 (*p* = 0.0344) F_2.384_ = 0.08 (*p* = 0.9224) F_2.384_ = 0.26 (*p* = 0.7670)	0.594 0.011 0.0004 0.001	1.000 0.562 0.062 0.092
	Control; *n* = 242	4.61 ± 1.59	4.35 ± 1.86	4.59 ± 1.63				
Agreeability scale	Addiction relapses (AR); *n* = 148	4.01 ± 1.73	4.66 ± 2.31	4.16 ± 2.12	Intercept AR/control rs6265 AR/control × rs6265	F_1.384_ = 465.25 (*p* < 0.0001) F_1.384_ = 7.33 (*p* = 0.0071) F_2.384_ = 0.27 (*p* = 0.7622) F_2.384_ = 0.12 (*p* = 0.8882)	0.548 0.019 0.001 0.0006	1.000 0.770 0.093 0.068
	Control; *n* = 242	5.43 ± 2.15	5.71 ± 2.33	5.38 ± 1.96				
Conscientiousness scale	Addiction relapses (AR); *n* = 148	5.43 ± 2.26	5.67 ± 1.15	5.39 ± 2.29	Intercept AR/control rs6265 AR/control × rs6265	F_1.384_ = 540.2 (*p* < 0.0001) F_1.384_ = 1.21 (*p* = 0.2728) F_2.384_ = 0.73 (*p* = 0.4798) F_2.384_ = 0.84 (*p* = 0.4343)	0.584 0.003 0.004 0.004	1.000 0.195 0.175 0.193
	Control; *n* = 242	5.67 ± 2.28	6.14 ± 2.51	6.31 ± 1.86				

*—significant result; AR—addiction relapses; M ± SD—mean ± standard deviation.

**Table 6 ijms-25-00788-t006:** Post hoc test (LSD) analysis of interactions between the patients in addiction treatment for the first time/control, the patients with addiction relapse/control and rs6265 *BDNF* and Openness scale and STAI trait scale.

**rs6265 *BDNF* and Openness Scale**						
	**{1}** **M = 3.97**	**{2}** **M = 5.58**	**{3}** **M = 4.67**	**{4}** **M = 4.59**	**{5}** **M = 4.61**	**{6}** **M = 4.36**
First time in addiction treatment G/G {1}		0.0000 *	0.5050	0.0698	0.0454 *	0.4793
First time in addiction treatment A/A {2}			0.3665	0.0001 *	0.0000 *	0.0134 *
First time in addiction treatment A/G {3}				0.9386	0.9540	0.7789
Control G/G {4}					0.9323	0.6439
Control A/A {5}						0.6048
Control A/G {6}						
**rs6265 *BDNF* and STAI trait scale**						
	**{1}** **M = 6.45**	**{2}** **M = 7.65**	**{3}** **M = 5.67**	**{4}** **M = 5.14**	**{5}** **M = 5.08**	**{6}** **M = 6.50**
Addiction relapses G/G {1}		0.0049 *	0.5647	0.0032 *	0.0010 *	0.9406
Addiction relapses A/A {2}			0.1336	0.0000 *	0.0000 *	0.0729
Addiction relapses A/G {3}				0.6923	0.6585	0.5623
Control G/G {4}					0.8532	0.0377 *
Control A/A {5}						0.0257 *
Control A/G {6}						

*—significant statistical differences, M—mean. LSD—least significant difference.

## Data Availability

Data are contained within the article.

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
