# Peer review of "The Relationship between the Brain-Derived Neurotrophic Factor Gene Polymorphism (Val66Met) and Substance Use Disorder and Relapse"

_ijms, 2024, doi:10.3390/ijms25020788_

Round 1
Reviewer 1 Report
Comments and Suggestions for Authors
To Authors:
The authors aimed to analyzed the association of the rs6265 polymorphism of the BDNF gene in a group of patients addicted to psychoactive substances participating in addiction treatment for the first time, in a group of post-relapse psychoactive substance abusers and a control group, resulting in the outcomes that statistically significant differences in the frequency of genotypes and alleles were found between all study groups, in which compared to the control, both study groups had statistically significantly higher scores for trait and state anxiety, and addicted patients in both groups also had higher scores on the Neuroticism and Openness scales and lower scores on the Extraversion and Agreeableness scales. The manuscript is largely well written and informative overall. However, there seem to be a few minor concerns in this manuscript. The paper will be improved when the authors revise them according to the following comments:
[Minor points]
1. Introduction:
“prefrontal cortex. cortex” should be changed to “prefrontal cortex”. [Page 2, Line 62]
2. Results:
If tables are numbered in the order of appearance, “Table 3” in Page 3, Line 96, should be exchanged for “Table 2” in Page 3, Line 101. [Page 3, Line 96; Page 3, Line 101]
2. Results:
If tables are numbered in the order of appearance, “Table 6” in Page 3, Line 123, should be exchanged for “Table 5” in Page 4, Line 140. [Page 3, Line 123; Page 4, Line 140]
Table 3:
It is unclear why some of the p-values are bold whereas others are not, since p-values for statistically significant results are not always written in bold letters.
Table 6:
“hjvu j kkkil” should be deleted if it means nothing.
4. Materials and Methods:
In the paragraph “4.4. Statistical Analysis”, the level of statistical significance should be clearly described, because it is unclear which results are statistically significant in the manuscript, although the word “significant” is often used in the manuscript. In addition, the authors should describe if they consider correction for multiple comparisons in results of analyses for many phenotypes including subscales in the present study.
Author Response
Dear Reviewer,
Thank you very much for your opinion on our manuscript and your valuable comments on the content and editorial aspects. All the comments have been discussed by the team. We fully agree with them. We have made corrections to the manuscript, and below - in our response to the reviews - we have included a description and explanation, as well as a precise indication of the line and verse where corrections have been made in the manuscript.
Yours sincerely
To Authors:
The authors aimed to analyzed the association of the rs6265 polymorphism of the BDNF gene in a group of patients addicted to psychoactive substances participating in addiction treatment for the first time, in a group of post-relapse psychoactive substance abusers and a control group, resulting in the outcomes that statistically significant differences in the frequency of genotypes and alleles were found between all study groups, in which compared to the control, both study groups had statistically significantly higher scores for trait and state anxiety, and addicted patients in both groups also had higher scores on the Neuroticism and Openness scales and lower scores on the Extraversion and Agreeableness scales. The manuscript is largely well written and informative overall. However, there seem to be a few minor concerns in this manuscript. The paper will be improved when the authors revise them according to the following comments:
- Introduction:
“prefrontal cortex. cortex” should be changed to “prefrontal cortex”. [Page 2, Line 62]
This is an editorial error, it has been corrected.
- Results:
If tables are numbered in the order of appearance, “Table 3” in Page 3, Line 96, should be exchanged for “Table 2” in Page 3, Line 101. [Page 3, Line 96; Page 3, Line 101]
Thank you for this comment. The place of tables in the manuscript has been changed; Table 2 - page 3, line 95-98, Table 3 - page 3-4, line 103 - 106.
- Results:
If tables are numbered in the order of appearance, “Table 6” in Page 3, Line 123, should be exchanged for “Table 5” in Page 4, Line 140. [Page 3, Line 123; Page 4, Line 140]
Thank you for this suggestion, but in this case, we would prefer to leave Table 6 here because our intention was to show the first ANOVA analyses for the two groups in Tables 4 and 5 and the pooled post hoc analysis in Table 6.
Table 3:
It is unclear why some of the p-values are bold whereas others are not, since p-values for statistically significant results are not always written in bold letters.
This editorial error; it has been corrected, and the statistically significant p-value is written in bold.
Table 6:
“hjvu j kkkil” should be deleted if it means nothing.
This is an editorial error. This has been removed.
- Materials and Methods:
In the paragraph “4.4. Statistical Analysis”, the level of statistical significance should be clearly described, because it is unclear which results are statistically significant in the manuscript, although the word “significant” is often used in the manuscript. In addition, the authors should describe if they consider correction for multiple comparisons in results of analyses for many phenotypes including subscales in the present study.
Thank you for this suggestion. In subsection 4.4 of statistical analyses, the following sentence has been added: Statistical significance was assumed at p < 0.05.
Reviewer 2 Report
Comments and Suggestions for Authors
The study explores the association of a well known coding polymorphism; rs6562, in the gene encoding BDNF between first time treatment and relapsed drug addicts and control groups.
They report a significant association between the G-allele and both rehabilitation groups (fist time and relapsed) compared to the non-addicted controls which is an observation consistent with previous multiple studies.
However, the study also includes personality traits such as agreeableness, neuroticism, anxiety and extraversion and found that individuals seeking rehab for the first time showed significant differences between both the control and relapsed rehab groups. Is there a possibility that these personality differences might be due to more extensive drug use and deprivation rather than the allelic variant? This needs to be discussed as a possible confounding factor.
I have a number of concerns about these conclusions. The first concern relates to the novelty of the study. Much of these studies have been carried out by many other groups in the past so the current study merely supports these other studies. Please emphasise where the study is novel.
The second concern rests with the low sample size used in the study (FTT=143 (1?), relapse=148, control= 242 which may produce a persuasive result for allele frequencies between each group but, when personality traits are also factored in, become less persuasive.
Thirdly, the error bars displayed in figures 1 and 2; particularly in the A/A genotype, seem to be too large to contribute to the conclusions.
Finally, little effort has been made in discussing the possible mechanisms involved or to highlight possible shortcomings of the study other than to say that "larger sample sizes are required".
In general, this is a nicely written study of an important subject but lacks originality or statistical power.
I would be happy to look over another draft if the authors are able to address my concerns.
Author Response
Dear Reviewer
Thank you for your comments and comments.
We re-analyzed the manuscript taking into account all the reviewers' comments. Below is a description of what was changed in the manuscript with lines marked. We also respond to individual comments, which turned out to be very useful, for which we thank you once again.
Comments and Suggestions for Authors
The study explores the association of a well known coding polymorphism; rs6562, in the gene encoding BDNF between first time treatment and relapsed drug addicts and control groups.
They report a significant association between the G-allele and both rehabilitation groups (fist time and relapsed) compared to the non-addicted controls which is an observation consistent with previous multiple studies.
However, the study also includes personality traits such as agreeableness, neuroticism, anxiety and extraversion and found that individuals seeking rehab for the first time showed significant differences between both the control and relapsed rehab groups. Is there a possibility that these personality differences might be due to more extensive drug use and deprivation rather than the allelic variant? This needs to be discussed as a possible confounding factor.
Thank you for this comment and for asking an important question.In our research, we try to take into account genetic, environmental and psychological factors.Differences were actually shown between the control group and the case group.Is this difference more due to the allelic record or to a factor related to the duration of substance use?We don't know that.However, we want to show a multivariate analysis.In this case, we agree with the reviewer's opinion that this should be discussed as a possible confounding factor.This comment was included in the manuscript:
Page14 , „Conclusions - “We emphasise that the analysis of genotypes and alleles in connection with personality-related factors is justified. Still, it should be remembered that it may also be a factor limiting the interpretation of the study.”
I have a number of concerns about these conclusions. The first concern relates to the novelty of the study. Much of these studies have been carried out by many other groups in the past so the current study merely supports these other studies. Please emphasise where the study is novel.
And here we agree with the reviewer. However, during the discussion on this point, divided tasks appeared in the research team. However, after discussion, we unanimously concluded that our study indeed supports other previous reports. BDNF is a gene studied in addictions, with particular interest in the rs 6562 variant. We have already conducted analyzes related to this polymorphism in various subgroups of addicts. However, the results are still not clear. In our research, we focus on multivariate analyzes and the separation of homogeneous subgroups. Through our many years of experience, we see that this is the right path. In our opinion, the innovative nature of the study consists mainly in multivariate analysis and the separation of subgroups. A short comment has been added in “Conclusions”. Page 14."However, the innovative nature of the study makes it necessary to create homogeneous subgroups - including those that take into account the personality characteristics of the respondents."
The second concern rests with the low sample size used in the study (FTT=143 (1?), relapse=148, control= 242 which may produce a persuasive result for allele frequencies between each group but, when personality traits are also factored in, become less persuasive.
Thank you for this comment. On the one hand, it can be said that this is a small group. On the other hand, the groups are carefully selected, these are patients from whom a detailed history of addiction was collected and examined by a specialist psychiatrist. Additionally, during the analysis, it was decided to divide the group into homogeneous subgroups regarding the moment of addiction. All patients stayed in a closed addiction treatment center and were recruited to the study after abstinence for at least three months. The criterion for inclusion in the group after at least three months of abstinence was related to conducting psychometric tests. Collecting such a group of addicts and testing them (so that they met the described criteria) took from several months to two years. It was also associated with withdrawal from therapy and sometimes even death of hospitalized patients. Similarly, the control group was very carefully selected. Not only matched according to age and gender but also examined by a specialist psychiatrist.
Thirdly, the error bars displayed in figures 1 and 2; particularly in the A/A genotype, seem to be too large to contribute to the conclusions.
Thank you for this comment. The conclusions have been corrected.
Finally, little effort has been made in discussing the possible mechanisms involved or to highlight possible shortcomings of the study other than to say that "larger sample sizes are required".
Thank you for your comment. Our main goal is to actually enlarge the group, although this is a difficult and time-consuming task. But we agree that we should have emphasized more strongly the weaknesses of the study - which we did in the conclusions.
In general, this is a nicely written study of an important subject but lacks originality or statistical power.
I would be happy to look over another draft if the authors are able to address my concerns.
Thank you again for your review and comments. They are all valuable to us. We have tried to improve the manuscript and draw conclusions more carefully, while not getting too excited about the research results.